# Structural Analysis of Low Defect Ammonothermally Grown GaN Wafers by Borrmann Effect X-ray Topography

**DOI:** 10.3390/ma14195472

**Published:** 2021-09-22

**Authors:** Lutz Kirste, Karolina Grabianska, Robert Kucharski, Tomasz Sochacki, Boleslaw Lucznik, Michal Bockowski

**Affiliations:** 1Fraunhofer Institute for Applied Solid State Physics (IAF), Tullastraße 72, 79108 Freiburg, Germany; 2Institute of High-Pressure Physics of the Polish Academy of Sciences, Sokołowska 29/37, 01-142 Warsaw, Poland; kgrabianska@unipress.waw.pl (K.G.); kucharski@unipress.waw.pl (R.K.); tsochacki@unipress.waw.pl (T.S.); bolo@unipress.waw.pl (B.L.); bocian@unipress.waw.pl (M.B.); 3Center for Integrated Research of Future Electronics, Institute of Materials and Systems for Sustainability, Nagoya University, C3-1 Furo-cho, Chikusa-ku, Nagoya 464-8603, Japan

**Keywords:** GaN, crystal growth, ammonothermal method, defects, X-ray topography, Borrmann effect

## Abstract

X-ray topography defect analysis of entire 1.8-inch GaN substrates, using the Borrmann effect, is presented in this paper. The GaN wafers were grown by the ammonothermal method. Borrmann effect topography of anomalous transmission could be applied due to the low defect density of the substrates. It was possible to trace the process and growth history of the GaN crystals in detail from their defect pattern imaged. Microscopic defects such as threading dislocations, but also macroscopic defects, for example dislocation clusters due to preparation insufficiency, traces of facet formation, growth bands, dislocation walls and dislocation bundles, were detected. Influences of seed crystal preparation and process parameters of crystal growth on the formation of the defects are discussed.

## 1. Introduction

Gallium nitride (GaN) wafers of the highest structural quality are needed for building optoelectronic and electronic device structures. These include laser diodes (LDs) as well as vertical field-effect transistors (FETs) and high electron mobility transistors (HEMTs) [1,2,3]. In order to prepare GaN wafers, bulk crystals have to be grown. Then, as a result of processing (wafering) procedures such as slicing, misorientation, drilling, grinding, lapping and mechanical polishing as well as chemo-mechanical polishing (CMP) one can obtain substrates from crystals. Today, three methods are applied for growing large in diameter GaN crystals: i. halide vapor phase epitaxy (HVPE) and its derivatives as oxide VPE (OVPE) or halide free VPE (HFVPE) [4,5,6]; ii. basic and acidic ammonothermal [7,8]; and iii. sodium flux [9]. Substrates, with (0001) crystallographic planes (i.e., *c*-planes) prepared to an epi-ready state, obtained from crystals grown by the first two methods are commercially available. However, GaN substrates of the highest structural quality are fabricated from crystals grown by the basic ammonothermal method [7,10]. These substrates are characterized by flatness of crystallographic planes (bowing radius higher than 15 m for 2 inch substrates), relatively low elastic deformation and threading dislocation density (TDD) at the level of 5 × 10^4^ cm^−2^ or even lower [11,12]. Ammonothermal GaN (Am-GaN) crystals can be highly conductive with free carrier concentration close to 10^19^ cm^−3^ and resistivity 10^−3^ Ω cm or semi-insulating (SI) with resistivity higher than 10^10^ Ω cm at room temperature (RT) [13].

Ammonothermal GaN crystals and wafers have low TDD, but obviously they are not free of defects. The knowledge about types, density and distribution of these defects is of high importance. Crystal growers need to know which GaN crystals are appropriate as seeds and which growth strategies will improve the material. Manufacturers of GaN-based devices are interested in the defect characteristics of GaN substrates, as defects can affect the performance, reliability and lifetime of the devices [14,15]. The types of defects in Am-GaN are quite diverse and their detection is difficult for various reasons. On one hand, there are µm-scale defects, such as threading dislocations (TDs). On the other hand, GaN crystals and wafers may have defects with lateral expansion on mm-scale. The latter group includes preparation-related defects or large defect clusters like dislocation bundles or dislocation walls related to inhomogeneous growth as will be shown in the course of this work.

There are several methods to observe and examine defects in crystals. However, the analysis techniques must be appropriate, in terms of density and size of the different defect types, for the material under investigation. In comparison to commercially available HVPE-GaN with typical TDD in the range of 10^6^–10^7^ cm^−2^, TDD of Am-GaN is two to three orders of magnitude lower. Conventional high resolution X-ray diffraction (HRXRD) analyses of the full width at half maximum (FWHM) prove to be unsuitable at a certain point, since the reflection broadening is in the range of the intrinsic resolution limits of the diffractometers and the sensitivity of this method is not sufficient to distinguish low-dislocation GaN crystals, with TDD below 10^4^ cm^−2^ and crystals with even lower defect density. Likewise, for such low TDD a method such as transmission electron microscopy (TEM) is limited or possible only with considerable effort. TEM techniques, albeit providing resolutions of the order of nanometers, cannot cover large areas of substrates, which is a prerequisite for wafer scale substrate analysis for material with a low defect density. Another characterization technique frequently used for GaN bulk crystals is defect selective etching (DSE) [16]. However, the method has the inherent disadvantage that it is strictly limited to defects that intercept the sample’s investigated surface and provides only very limited information about the arrangement of the defects in the volume of the crystal. Other methods include cathodoluminescence, photoluminescence and Raman spectroscopy topography. Defect characterization is in principle possible with all the mentioned analysis techniques. These methods are however not widely used for a routine inspection of crystals or wafers with TDD level as in Am-GaN as they do not cover the field of view simultaneously from µm to mm scale and are either laborious, destructive or time-consuming.

A more suitable technique for defect analysis of Am-GaN crystals and substrates is X-ray topography (XRT). In this work, the variant of Borrmann effect XRT is used. We show that this laboratory technique in transmission geometry is well suited for a routine analysis of low defect Am-GaN wafers for the development as well as process monitoring. The Borrmann effect, or anomalous transmission, occurs in XRT of highly absorbing crystals with very low normal transmission either because the crystals include heavy elements or because they are very thick (typically crystals with µt > 5 for the wavelength used, with µ = normal linear absorption coefficient and t = thickness of the crystal) [17,18].Under these conditions of a high µt, the waves of the incident beam propagating through the crystal outside the perfect-crystal reflection range are practically completely absorbed. Therefore, the kinematic contrast (“extinction contrast”) caused by the distorted regions around defects, such as dislocations, does not occur. However, defect imaging is possible in crystals that exhibit a pronounced Borrmann effect, i.e., a strong reduction in absorption for waves that are within the perfect crystal reflection range. The minimum of absorption arises for those incident waves creating in the crystal wave fields with energy flux parallel to the reflecting lattice planes [17]. This effect of anomalous transmission depends strongly on the perfection of the crystal. Defects of the crystal lattice, such as dislocations, reduce the effect and lead to a locally reduced intensity in both the reflected and in the forward transmitted beam. The resulting contrast is often referred to as the “Borrmann contrast”. Contrast effects due to the curvature of the wavefield rays in the deformed regions around the defects can be implied leading to an increase in intensity in some parts of the defect image [17]. A more detailed description of the Borrmann effect can be found in the literature, e.g., [18,19,20,21,22,23].

So far, XRT using the Borrmann effect has been reported mainly for highly absorbing crystals of semiconductors such as Ge [24], GaAs [25,26], CdTe [27] and ZnGeP_2_ [28,29,30], as well as the electro-optical material KH_2_PO_4_ [17]. In this paper we apply this method for three 1.8 inch highly conductive Am-GaN substrates. As far as we know, there are no reports of the Borrmann effect for GaN in the literature as yet. This study reveals that not only micro defects, such as dislocations and clusters of macro defects can be analyzed, but also the history of the grown crystals used for preparing these wafers can be traced in detail on the basis of the observed defect patterns. This provides important information on Am-GaN crystals that crystallize in a specific way from slender seeds in many regrowth runs and with applying tiling technology [10]. It should be noted that the measurements and analysis presented in this paper are ineffective when applied to commercially available HVPE-GaN, typically prepared with a foreign seed due to the low structural quality of this material and massive lattice distortion occurring in it. The observation of the Borrmann effect in the investigated Am-GaN wafers proves the high structural quality of the material.

## 2. Experimental

### 2.1. GaN Crystal Growth and Wafer Preparation

Basic ammonothermal crystal growth technology, described in detail elsewhere [7,10,31], is based on dissolving polycrystalline GaN feedstock in one zone of a high-pressure reactor filled with supercritical ammonia and transporting the dissolved material to the second zone, where the solution is supersaturated and the crystallization of GaN takes place on native seeds. A negative temperature coefficient of solubility is observed in the ammonobasic approach [32]. As a consequence of retrograde solubility, the chemical transport of GaN is directed from the low-temperature solubility zone to the high-temperature crystallization zone. At the beginning of the growth process, the feedstock zone is heated, and the material placed there starts to dissolve in ammonia. Herein, the feedstock temperature is higher than the temperature of the crystal growth zone. At dissolution stage a back etching process occurs; “the seeds are coupling with the solution”. Then, the crystal growth zone is heated to temperature higher than that in the feedstock zone. The temperature of the feedstock zone decreases, and the crystal growth run starts. The seeds are placed on special metal holders that are hung on different levels in the crystal growth zone of the reactor. The primary seeds are shaped like long slender sticks. They are overgrown in the lateral 
$‹11\bar{2}0›$
 directions during many crystallization processes (due to the slender shape of the seed sticks mainly in the 
$\left[11\bar{2}0\right]$
 and 
$\left[\bar{1}\bar{1}20\right]$
 directions). After reaching the desired lateral size, the crystals are joined in pairs by tiling in order to obtain a larger crystal. To multiply lateral overgrowth and tiling requires continuous mechanical processing of the crystals in the stage of their preparation for a given growth run. The crystals are mainly trimmed and lapped in a proper way. Figure 1 represents a scheme of the multi regrowth of a slender GaN seed and a scheme of combining two newly-grown, already separated from their seeds, crystals by tiling technology. When the combined crystals reach the correct size by growth in the lateral 
$‹11\bar{2}0›$
 as well as the vertical 
$\left[000\bar{1}\right]$
 crystallographic directions (growth in the latter direction is not marked), they are used for preparing substrates, shown in Figure 1. As already mentioned, then the wafering procedures are applied.

For the purpose of this work three highly conductive (n∼10^19^ cm^−3^) Am-GaN crystals were randomly selected for preparing three 1.8-inch substrates: A, B and C. In order to properly analyze the wafers with XRT in transmission geometry both surfaces, the (0001) and 
$\left(000\bar{1}\right)$
 crystallographic planes, were prepared to the epi-ready state.

### 2.2. Structural Analysis of the GaN Wafers

A XRTmicron laboratory X-ray topography camera (RIGAKU, Tokyo, Japan) was used for the imaging of the Am-GaN 
$\left(0001\right)$
 wafers. The camera is equipped with a high brilliance microfocus X-ray source combined with a multilayer X-ray optics. For the imaging, Cu-Kα_1_ radiation (λ = 154.06 pm, 8.05 keV) was used. With a GaN wafer thickness of about 450 µm, the µt criterion amounts to about 13.6 for the used Cu-Kα_1_ radiation (µ [cm^−1^] is the linear X-ray absorption coefficient and t [cm] is the crystal thickness). This meets the conditions for X-ray topography using the Borrmann effect if the Am-GaN wafers have sufficient perfection. At first, overview images of the entire wafers were performed in projection transmission geometry, the so-called Lang technique [33]. Imaging of each wafer was performed by exposures using type 
$11\bar{2}0$
 reflections, namely 
$11\bar{2}0$
, 
$\bar{1}\bar{1}20$
, 
$1\bar{2}10$
, 
$\bar{1}2\bar{1}0$
, 
$2\bar{1}\bar{1}0$
 and 
$\bar{2}110$
. In order to make a correction of curvature of the Am-GaN wafers, an automated Bragg angle control was used to keep the crystals in a scattering condition during the scans [34]. The X-ray topographs were recorded by a high-resolution CCD camera (5.4 μm × 5.4 μm pixel size). The scanning speed was 1 mm per minute. In addition to the overview images, higher resolution images of individual regions of interest were taken in snapshot mode with an ultrahigh resolution CCD camera (2.4 μm × 2.4 μm pixel size) e.g., to identify details of the defects.

## 3. Results

Figure 2a–c present XRT topographs of the three examined substrates using type 
$11\bar{2}0$
 reflections. On all topographs, higher X-ray intensity appears by increased blackening. Under the measurement conditions outlined above, the defects appear by Borrmann contrast, i.e., by locally reduced intensity. For all the samples, contrast of significantly different defects can be observed on µm scale up to mm scale. Moreover, the defects have an inhomogeneous distribution and density. The µm scale defects, hereafter also referred to as microscopic defects, are typically threading dislocations (TDs) of the screw, edge, or mixed type; on the other hand, mm scale defects, hereafter called macroscopic defects, are extended defects or defect clusters (e.g., defects related to tiling seams or subsurface damage, traces of facets, growth bands, different forms of dislocation bundles and dislocation walls). The mentioned types of microscopic and macroscopic defects are found in all three investigated wafers. Using wafer B as an example, such macroscopic defects are marked in Figure 3.

Below, all types of defects detected by Borrmann effect XRT in the investigated three Am-GaN substrates are analyzed and discussed in detail.

### 3.1. Microscopic Defects

An enlarged topograph, 
$1\bar{2}10$
 reflection, of wafer C with an array of dislocations exhibiting Borrmann contrasts is shown in Figure 4. Each dislocation is visible as a many-lobed bright and dark contrast forming a rosette. Defects on the microscopic scale in Am-GaN are mainly threading dislocations (TDs) propagating nearly along the *c*-axis as was already shown by different studies using e.g., defect selective etching analysis or synchrotron XRT techniques [11,12,35,36]. Rosette contrasts of TDs in GaN, however, are reported here for the first time. The directions of the bright and dark lobes of the rosettes are partly different, indicating that identical as well as different types of TDs are present. According to their Burgers vector *b*, three different types of TDs are observed in GaN crystals, namely screw-type (TSD, *b* = n*c*), edge type (TED, *b* = m*a*) and mixed-type (TMD, *b* = n*c* + m*a*; with *a*, *c* = lattice parameters and n, m = 1, 2, etc.) [12,35,36,37]. At the present stage of investigation, we are not yet able to assign the rosette contrasts to an exact TD type and this will be the subject of further research. However, the bright-dark contrasts appear to be formed by areas of compression and tension of the GaN crystal lattice, and a comparison of these experimental rosettes with XRT image contrast theory suggests that they are formed by the microstrain of dislocations with an edge type dislocation component, either pure TEDs, or TMDs with an edge component. This observation seems plausible since, according to the literature, 96–97% of TDs in Am-GaN are of the mixed type [12,36] and the rosette contrasts are therefore probably generated by predominantly TMDs. Considering the different possible combinations of *c* as well as *a* components in TMDs, with *c* components of opposite signs and six different types of *a* components, 12 combinations of TMDs are possible [37]. So, a different appearance of the rosette contrasts can therefore be expected. Similar rosette contrasts by Borrmann effect XRT have been reported for TEDs in GaAs and ZnGeP_2_ [25,29,30].

The threading dislocation density (TDD) in the three measured samples varies locally considerably. Figure 5 presents dislocation regions existing in the analyzed samples. Four areas were exemplary distinguished with high TDD on the order of ~5 × 10^5^ cm^−2^ (Figure 5a); medium TDD (2 × 10^5^ cm^−2^; Figure 5b), low TDD (4 × 10^4^ cm^−2^; Figure 5c), and ultra-low TDD (3 × 10^3^ cm^−2^; Figure 5d).

An interesting observation is that there are often relatively sharp demarcations in terms of the density of TDs and there appears to be a correlation with the individual steps of multi regrowth of slender GaN seeds. Areas of equal TDD occur in bar-like structures. The bars are perpendicular to one of the fastest growth directions, e.g., the 
$\left[11\bar{2}0\right]$
 directions and its opposite direction 
$\left[\bar{1}\bar{1}20\right]$
, but initially not to the four symmetrically equivalent 
$\left[2\bar{1}\bar{1}0\right]$
, 
$\left[\bar{2}110\right]$
, 
$\left[1\bar{2}10\right]$
 and 
$\left[\bar{1}2\bar{1}0\right]$
 directions. This effect is related to the slender shape of the seed sticks, as described in Section 2.1. The width of the bars corresponds to the increase in the lateral crystal area of each regrowth step for the seed broadening. The trend objective is that with each regrowth step in the outward bands, TDD decreases with a relatively sharp boundary compared to the neighboring region. i.e., the TDD decreases stepwise from the center of the wafer with typical TDD of about ~5 × 10^5^ cm^−2^ down to TDD of low 10^3^ cm^−2^ at the wafer edge. While in the first seed enlargement steps growth was limited to only two directions, in the later course seed enlargement typically occurred in all type 
$‹11\bar{2}0›$
 directions and areas with ultra-low TDD emerged at the wafer edge in all these six possible directions. This defect or growth pattern with the bar-like structures of the individual regrowth steps and the decrease in TDD towards the edges can be seen particularly clearly in the overview image of substrate B (Figure 2b). The dislocation-poor areas at the edges are characterized by a medium-dark, homogeneous contrast due to a distinctive Borrmann effect.

### 3.2. Macroscopic Defects

The TDs described in the previous section form a kind of basic matrix at the microscopic level. Locally, these defects are accompanied by the macroscopic defects which are discussed in more detail below.

#### 3.2.1. Defects Related to Seed/Crystal Preparation

*Tiling seams*—In all three substrates, very distinctive line-shaped contrasts of defects running across the entire wafer and quasi dividing the substrates into two halves can be observed (see Figure 2). As an example, Figure 6a shows these contrasts imaged by XRT for substrate B; Figure 6b shows a highly enlarged part of the tiling seam. The identification, as well as the origin of these defects is simple: by combining two seed crystals to obtain a larger one, a tiling seam is formed (see the scheme presented in Figure 1, step 3). After examining the tiling seam in detail, one can observe that the seam does not show contrasts of a single line, but several almost parallel lines. In the case of substrate B, six lines of contrast running parallel to the 
$\left[\bar{1}100\right]$
 direction are observed. Four of the contrast lines appear bright and two of the contrast lines appear dark. This effect is caused by a slight bending of the basal plane at the tiling seam as it forms a grain boundary. The overall pattern of line structures can be described as chains of threading dislocation bundles. The main course in the depth of the dislocation bundles is along the ‹0001› direction. The special characteristic of these dislocation bundles is that they are arranged in chains and their exit points on the surface appear like closed line structures due to their high density. In addition, the high density of TDs arranged as chain-like dislocation bundles also prevents the observation of rosette contrasts with detailed resolved lobes, which is the case for single spatially isolated TDs as described in Section 3.1. The defective crystal region caused by the tiling seam (or grain boundary) appears surprisingly wide with ~2 mm lateral extent, considering that the two seed crystal halves were in contact during growth. The reason for this is that the respective boundary crystal edge has been obliquely flattened and therefore the defect area has widened. For a better understanding, an optical image after photo-electrochemical etching (Figure A1) of such a tiling seam area in the cross section of a similar Am-GaN crystal (not from this study) is shown in Appendix A.

*Defects caused by subsurface damage*—Another type of defect related to crystal preparation is caused by subsurface damage due to slicing and/or polishing procedures. The defects formed during the mentioned above seed crystal preparations were not completely removed, neither in the mechanical, chemo-mechanical nor in the etching back process, and remained as subsurface damage. The continuation of crystal growth on these areas inevitably leads to the formation of defects. Figure 2b and Figure 3 show such defects on the example of substrate B. The white contrasts of the defects are present across the almost complete substrate. These contrasts run across both halves of the seeds, which were connected by the tiling technique to form a larger crystal. No doubt, these defects appeared after the tiling step. This hypothesis is supported by the fact that the end of this defect pattern is limited in the 
$\left[11\bar{2}0\right]$
 and 
$\left[\bar{1}\bar{1}20\right]$
 direction by areas that were formed in a subsequent generation of the crystal growth process. As can be identified in an enlarged section of this defect cluster (Figure 7), the distribution of contrasts in some areas is reminiscent of the traces of a saw or a polishing tool. The exact interpretation of the white contrasts is, however, difficult. Most likely they are networks of high-density TDs induced by subsurface damage in the seed crystal. At these locations with high dislocation density, the Borrmann effect is nearly extinct and the transmission of the X-rays is practically completely blocked.

*Facets*—Next defects, well visible by XRT, are associated with facets (or hillocks). The exact reason for the formation of this type of defect is not clear. One possibility is that these defects were formed due to improperly prepared seed surfaces with excessive roughness of the 
$\left(000\bar{1}\right)$
 Am-GaN planes. A consequence of surface roughness of the seed is 3D growth that causes the formation of hillocks. In the ongoing growth process, this may lead to the propagation of macrosteps flowing along the slopes of the hillocks. Over time, some hillocks grow out into facets and begin to dominate. Another possibility could be that inhomogeneities in the crystallization process in the autoclave, such as non-uniform temperature distribution and/or non-uniform transport of reagents to the seed crystals, especially at the beginning of the growth run, are responsible for the facet formation. A step bunching effect, due to constitutional supercooling, can take place and trigger the formation of hillocks. However, it can also not be ruled out, that the facet formation in Am-GaN is a combination of both causes, improper seed preparation and growth inhomogeneities. The facets typically become visible to the naked eye after growth on the surface of 
$\left(000\bar{1}\right)$
 growth planes [11], but traces of facets are also visible as plane defects in the XRT images after wafering. Therefore, the facets that occurred during growth can be clearly traced retrospectively in every sliced Am-GaN wafer and they can be observed in all three wafers examined (Figure 2). It should be made clear that in the topographs the facets themselves are not visible, but the traces where they originated in the crystal. For this reason, these planar defects should be termed traces of facets. The appearance of this defect type in the topographs changes as a function of the diffraction vector, as shown in the example of a former facet region from wafer A (Figure 8). The identical area with traces of a facet was recorded with the reflections 
$\bar{1}\bar{1}20$
, 
$\bar{1}2\bar{1}0$
, and 
$2\bar{1}\bar{1}0$
, respectively. For the reflection 
$\bar{1}\bar{1}20$
, the planar defect is almost extinguished and only very faint shadows are visible (Figure 8b). For the reflections 
$\bar{1}2\bar{1}0$
 and 
$2\bar{1}\bar{1}0$
 (Figure 8a,c), the area of the former facet is clearly visible as a triangular contrast. The demarcation of the contrast to the surrounding area is formed for the reflection 
$\bar{1}2\bar{1}0$
 at the upper edge by a dark line, at the lower edge by a bright line. For the reflection 
$2\bar{1}\bar{1}0$
 it is the other way round. The contrast generated by these planar defects is caused by a slight bending of the reflecting planes along the boundaries and results from a small difference of the lattice parameters in adjacent regions. The local difference in the lattice parameters occurs due to a different incorporation of dopants at the vertical 
$\left(000\bar{1}\right)$
 growth front compared to the inclined pyramidal growth directions of the facets.

#### 3.2.2. Defects Related to Growth

*Growth bands*—Figure 9 shows two topographs of an area of substrate A imaged with diffraction vectors *g* pointing in opposite directions, namely 
$\bar{1}\bar{1}20$
 (Figure 9a) and 
$11\bar{2}0$
 (Figure 9b). The growth front of the Am-GaN crystal proceeded from right to left in 
$\left[11\bar{2}0\right]$
 direction. In Figure 9a is a band of dark contrast is visible and in Figure 9b a band of a bright contrast can be seen. These contrasts are growth bands. After Klapper [38], growth bands are commonly formed in crystals grown from melts and solutions containing impurities, deviations from stoichiometry or constituents of the solvent may contribute to these imperfections. Growth bands (micro-inhomogeneities) are associated with fluctuations of growth conditions (e.g., heat and mass transfer, back-etching/regrowth and other growth discontinuities) and thus are arranged as bands normal to the growth direction. The growth bands may occur isolated or in a dense sequence (striations) [38]. In the Am-GaN crystals studied here, these growth bands are typically found in the boundary regions of the stages of multi-regrowth starting from slender GaN and continuing through the individual steps of lateral seed extension. Therefore, these growth bands could be the consequences of the back-etching or regrowth processes. As can be seen by comparing Figure 9a,b, the dark-bright contrast of these planar defects is reversed if opposite diffraction vectors are used. In this case they are 
$\bar{1}\bar{1}20$
 (Figure 9a) and 
$11\bar{2}0$
 (Figure 9b). This effect is due to a slight bending of the reflecting planes along the boundaries and results from a small difference of the lattice parameters in adjacent areas [17,38]. It should be kept in mind that not only in the lateral direction, but also in the vertical 
$\left[000\bar{1}\right]$
 growth direction, phenomena such as growth bands and striations are typically present in ammonothermally grown GaN due to the regrowth steps, as was already shown by Sintonen et al. [39]. Another observation that can be made in the section of the two topographs (Figure 9) is that TDD gradually decreases in the lateral 
$\left[11\bar{2}0\right]$
 growth direction; to the right of the growth band TDD is the highest, in the growth band area itself there is a medium TDD and to the left of the growth band TDD is lowest. This effect has already been explained above in Section 3.1.

*Dislocation walls*—In Figure 10 (
$\bar{1}2\bar{1}0$
 reflection XRT image of a part of substrate A), again a white contrast band can be seen; however, these contrasts are dislocations. The density of dislocations is extremely high and individual dislocations are barely resolved. These dislocations appear like a network forming a continuous wall that extends perpendicularly to the 
$\left[\bar{1}2\bar{1}0\right]$
 direction of growth. Due to the high defect density, the Borrmann effect is almost absent for this area. An exact reason for the formation of this dislocation network is difficult to establish and only speculations can be made in this regard. On the left, in front of the wall of dislocations, there are dark contrasts which indicate growth bands, similar to those growth bands of the example in Figure 9. A plausible explanation could be that the magnitude of the composition fluctuation in the growth bands was so large that the elastic stress that occurred was relieved by misfit dislocation formation [40]. Again, as with the example before (Figure 9), separated areas of distinct TDD can be observed for the different lateral regrowth areas with the lowest TDD in the regrowth area at the outer edge in the 
$\left[\bar{1}2\bar{1}0\right]$
 direction.

*Dislocation bundles*—Besides the TDs that form a kind of a “background matrix”, described in Section 3.1, another category of TDs is observed in the investigated Am-GaN wafers, namely dislocation bundles (DBs). The DBs are TDs that arise locally as clusters in high density. Individual TDs are almost not resolved in these clusters using XRT. However, distinct bright and dark contrasts are observed, some forming large many-lobed contrast rosettes, comparable to the rosettes of the single TDs, but much larger in their dimen-sions. The DBs occur in quite different forms. There are: single DBs, chains of DBs, as well as special arrangement of DBs, called by us “honeycomb” and “cauliflower”. Figure 11 shows single DBs of wafer A (Figure 11a), wafer C (Figure 11b), and wafer B (Figure 11c), respectively. The many-lobed contrast rosettes of the DBs are each embedded in a matrix of TDs and one can clearly see their large size compared to individual TDs. The occurrence of DBs seems to be independent of the density of TDs in the matrix, because as the series of Figure 11 shows, there are DBs in Am-GaN crystal regions with higher TDD (Figure 11a) intermediate TDD (Figure 11b), and very low TDD (Figure 11c).

In addition to single DBs, also accumulations of DBs can arise in chain-like arrangements (Figure 12). The DB chains always run along an *a*-plane 
$\left\{11\bar{2}0\right\}$
 growth facet. In some cases, the DB chains are interrupted with small regions without DBs (Figure 12a, part of substrate A) or the chains are completely continuous (Figure 12b, part of substrate C).

Another and quite striking form of DBs are clusters consisting of six individual DBs with a hexagonal arrangement (Figure 13a–c). For a better visualization, identical images are additionally shown underneath with a hexagon as an orientation guide for the eye, which connects the individual DBs of these defect clusters. The outer edges of the hexagons always run along the *a*-plane 
$\left\{11\bar{2}0\right\}$
 prismatic facets and these clusters can reach the size of more than 1 mm edge length for the wafers investigated here (and more than 2 mm edge length for substrates examined in other studies). Due to their appearance, these DB clusters are referred to as “honeycomb defects” in the following part of this paper. To the best of our knowledge, this type of defect has so far not been described for GaN, nor for any other type of crystal. Figure 13a shows a perfect honeycomb of six DBs in a matrix with medium TDD (part of wafer C). The dynamic rosette-shaped bright and dark contrasts of the honeycombs are particularly clearly visible in Figure 13b observed in an area of substrate C with extremely low TDD. The honeycomb of this example has two small errors. Two more DBs, marked by stars, can be seen and they almost fit into the shape of the honeycomb. However, the honeycomb clusters can also arise with strong distortions of the GaN crystal lattice. Figure 13c shows such an example (part of substrate C). An interesting aspect is that the hexagonal shape of the honeycomb is still recognizable, although the crystal lattice inside the honeycomb is distorted in such a way that the Borrmann effect in this region is almost extinguished. The shape of this defect cluster is reminiscent of a “cauliflower” and is therefore correspondingly called as “cauliflower defect”. However, in principle this defect is only a variant of the honeycomb defect with a strong lattice distortion.

The single and chain-like DBs are defects that are manifested in the growth steps of seed crystal enlargement and always arrange themselves along *a*-plane 
$\left\{11\bar{2}0\right\}$
 facets. It is shown that these DBs arise not only at the interface between two regions of different regrowth steps (as in the example of a DB chain visible in Figure 12b), but they also can arise suddenly in the middle of a seed enlargement growth phase (as in all examples shown for single DBs (Figure 11) and the DB chain of Figure 12b). The formation point of the DBs with honeycomb structure (Figure 13) is currently less clear. It could be during the growth for seed crystal enlargement along *a*-plane 
$\left\{11\bar{2}0\right\}$
 facets or at the growth in the vertical 
$\left[000\bar{1}\right]$
 directions. Likewise, it is difficult at this point in time to name a cause for the arising of all of these defect clusters and it requires further targeted investigation. A particularly interesting variant of these defects is the honeycomb defect, which formation seems to be related to the hexagonal symmetry of the wurtzite lattice and might cause a kind of energy minimization for the strain budget. A detailed discussion of these defect clusters will be given elsewhere [41].

## 4. Discussion and Conclusions

In this study it has been demonstrated that Borrmann effect XRT is an excellent technique for the analysis of strongly absorbing GaN substrates or crystals. An essential prerequisite for the successful application of this technique is a high degree of structural perfection of the crystals in order to obtain the Borrmann effect. This condition was met by the Am-GaN substrates studied here. The Borrmann effect topography can be a useful addition to the portfolio of XRT techniques for the defect investigation of GaN crystals of high structural perfection since it enables the observation of more refined and subtle contrast phenomena. As an example, synchrotron white-beam XRT or monochromatic XRT measurements in the back-reflection geometry were often used for the analyses of TDs in Am-GaN (0001) wafers [12,35,36,37]. With these measurement techniques, the TDs generate bright-dark punctual contrasts with shadings. Especially for threading edge (TEDs) and threading mixed dislocations (TMDs), these punctual contrasts are often very small, weak, as well as diffuse, so that an identification of the type and direction of the TDs is often difficult or impossible at all. As shown in this work, strong and filigree bright-dark rosette contrasts of the TDs can be observed with Borrmann effect XRT for Am-GaN. These distinct and clear contrast structures should make it possible, e.g., by comparison with simulation of the intensity contrast, to determine TDs with respect to their type and orientation. Another advantage of Borrmann effect XRT is that it can be performed as a laboratory method that allows a quick examination of the structural quality of entire wafers or even wafer series. This enables fast feedback for crystal growers or technologists.

When evaluating the obtained topographs for all three Am-GaN wafers (Figure 2 and Figure 3), it is possible to trace in detail the history of the crystallization and preparation process. For many defects, their origin is obvious and clearly correlated to the crystallization and preparation process; however, there are also defects which origin is still unclear. Additional analysis is needed for explaining how such defects are created. In general, a fundamental prerequisite for the successful growth of a crystal with a low defect density is a suitable seed. A proper preparation of the seed crystal is an extremely sensitive process, the result of which is immediately visible in the grown crystal. In addition to the seed preparation, the crystal growth process itself can trigger the appearance of crystal defects. The cause can occur during the different process stages such as heat-up, back-etching of the seeds, temperature transition from back-etching of seeds to actual growth process or cooling-down. Last but not least, wafering can also create defects, e.g., scratches not completely polished away or subsurface damage due to insufficient chemo-mechanical polishing, which may then have a negative effect on the epitaxial growth for the deposition of the device structure.

A very striking observed macroscopic defect type, that appears due to seed preparation, is associated with the tiling technology. Dislocation bundles with high density and linearly bunched extend across the entire wafers (Figure 6). An interesting observation here is that the tiling seams are not formed from a single chain of dislocation bundles, but as in the example shown in Figure 6, several, in this case six, chains of dislocation bundles occur. These aligned chains of dislocation bundles form a kind of “sub-grain boundary” and the threading dislocations of the bundles, which compensate the stress, are therefore similar to “geometrically necessary dislocations”. Obviously, by means of the tiling technique, the crystals’ structural quality must be locally reduced. On the other hand, this technique offers the possibility to obtain larger crystals in the fastest way. During tiling, it is impossible to align the seed crystals in such a way that their crystal lattices can grow together without misorientation and thus without defects. Figure 14 shows a scheme of the tiling technique. The three misorientation angles: α, β and γ, are marked. However, proper mutual alignment of the seeds in three planes allows to obtain crystals in which the angular misalignment of α, β, and γ is below 0.02 degrees, as was confirmed by HRXRD measurements.

Surface preparation is a complex multi-step process and includes steps such as cutting, mechanical, chemo-mechanical polishing and the back etching in the autoclave for growth. The origins of many possible defects are initiated in this multi-step process. Subsurface damage in seed crystals, e.g., in the case of insufficient surface preparation after cutting and/or polishing, can lead to the formation of defect patterns. An example on macroscopic scale, that could be clearly assigned to subsurface damage, was shown in Figure 2b and Figure 7. The traces of a saw or a polishing tool have initiated the formation of dislocation structures that extend over large areas of substrate B. Another type of defects, initiated by improperly performed wafering procedures on the 
$\left(000\bar{1}\right)$
 Am-GaN surface or/and inhomogeneities in the crystallization process are traces of facets. After wafering, the regions of the former hillocks become visible in XRT as planar defects in the substrate slices (Figure 2 and Figure 8). These planar defects are formed due to a small difference of lattice parameters compared to the adjacent crystal regions and result in slight bending of the reflecting planes along the boundaries. In the ideal crystallization process, only one hillock would be present in the center of the growing crystal.

In addition to the types of defects at the macroscopic level, however, there are also defects at the microscopic level that are possibly related to improper surface preparation. As was recently shown by Grabianska et al. [11], using optical birefringence measurements for the analysis of the stress induced polarization effect (SIPE), internal stress is present in Am-GaN wafers. The internal stress correlates with TDD, as was shown by etch pit density analysis in this study [11]. The typical distribution of TDs in Am-GaN wafers found by Grabianska et al. could be confirmed in this work by means of Borrmann effect XRT. The TDs are formed most probably at the beginning of the growth, just after the back etching process. The easiest way to explain this is to imagine the growth of many islands with well-formed semi-polar and nonpolar facets. Before their coalescence, the incorporation of dopants on the facets is different than on the 
$\left(000\bar{1}\right)$
 plane. When a full coalescence happens, TDs are generated. It should be remarked that a 1–2 orders of magnitude lower TD is found in the most recent grown areas in the lateral 
$‹11\bar{2}0›$
 direction. The question that should be posed is why the growth starts with a formation of 3D islands? The first possible reason could be associated with the surface preparation of the seed. The 
$\left(000\bar{1}\right)$
 surfaces are not sufficiently prepared and are therefore macroscopically nonuniform. If back etching is applied in order to dissolve the seeds in the solution, this process does not lead to an atomically flat surface. Similar to the growth process, the etching is also anisotropic. However, another or further explanation could be, with an inhomogeneous convective flow and from this, inhomogeneous supersaturation and reactants flow on the surface of the growing crystal could occur. Both the seeds’ surface preparation for growth as well as the convective flow need to be analyzed and studied in the future.

In the overview topographs (Figure 2 and Figure 3), one can clearly see the individual interfaces of the steps of multiple regrowth of lateral Am-GaN for seed enlargement. The reason for the visibility is that macrodefects such as growth bands (Figure 9) and, in some cases, additional dense dislocation networks that arise as extended dislocation walls (Figure 10) are formed. It is obvious that the beginning of the crystallization run (initial nucleation) did not proceed in the desired way. The discussed defects are related to the crossover appearing during the ammonothermal process: from the back-etching of the seeds at lower temperature, where the coupling of the solution with the seeds takes place, to the beginning of GaN growth stage. In this crossover a temperature transmission is performed in the autoclave [10,11]. As described, since basic ammonothermal GaN growth is a process with retrograde solubility, the temperature in the area of the autoclave for the feedstock is reduced in this step and, conversely, the temperature in the area of the seed crystals is increased. Temperature control during this transition is critical. If this control is not successful, the result can be a non-uniform incorporation of gallium or nitrogen or an increased incorporation of impurities at the growth front. This deviation in the local composition of the crystal, and thus also the local change in the lattice parameters, leads to a slight bending of the crystal plane and, therefore, this effect becomes visible as growth bands in the XRT images. The magnitude of the composition fluctuation in the growth bands can be so large that the elastic stress that occurs is relieved by misfit dislocation. In an extreme case, constitutional supercooling effects may be observed, which generate detrimental grain boundary structures with high dislocation density. 

Another type of defects that have not yet been described in the literature for Am-GaN are dislocation bundles (DBs), which occur in various forms, namely single DBs (Figure 11), chains of DBs (Figure 12), and DBs in a hexagonal arrangement (the honeycomb defect, Figure 13). The DBs show up in Borrmann effect XRT as large many-lobed contrast rosettes. The honeycomb defect is particularly striking. This defect cluster reaches expansions of over 2 mm. However, an arrangement of the defects with hexagonal symmetry seems to have energetic advantages for the growing crystal. The origin of the DBs, and especially the honeycomb defect, is still unclear. Presumptions are that this defect type is related to inhomogeneities of supersaturation at the growth front. A crucial point is the control and a proper distribution of the temperature since the temperature determines the convective flow and thus the material transport to the growth front. An inhomogeneous material transport influences the supersaturation at the growth front. Nonuniform and non-controlled supersaturation can be the origin for the formation of defects in the growing crystal. Clearly measurable fluctuations in the autoclave of fluid temperature were observed in the experiments with supercritical ammonia for Am-GaN growth by Schimmel et.al. [42]. Further studies are needed to understand the formation of this type of defect.

It has to be remarked that it was possible to detect all the described defects in Am-GaN by Borrmann effect XRT due to extremely low defect density in this material. Interestingly, HVPE-GaN grown on foreign substrates looks much more homogeneous, e.g., in XRT techniques that are used in back-reflection geometry. However, this result is misleading, since the defects present in this material cannot be resolved properly with XRT. This is due to TDD 2–3 orders of magnitude larger than in Am-GaN. One can say that this homogeneity concerns the high dislocation density of the HVPE material grown on foreign seeds. It was shown that in commercially available HVPE-GaN the dislocation rearrangement into cell networks can take place under external or internal stress in the course of plastic relaxation [43]. The Borrmann effect can therefore hardly be observed in HVPE-GaN grown on foreign substrates due to the low material quality; it is nearly extinct. The same applies for ammonothermally grown GaN crystals, which also use a foreign substrates as seeds [44]. Examples of unsuccessful Borrmann effect XRT for HVPE and amonothermal GaN, grown with foreign substrate seeds, are given in Appendix B (Figure A2).

We believe that the presented results in this paper show a new level of quality of the Am-GaN crystals. The influence of some of the new defect types found in the Am-GaN substrates on the electrical properties of devices is still unclear and needs to be studied in the future. However, there is no doubt that the use of these crystals as substrates for the realization of GaN-based optoelectronic and electronic device structures will enable significant progress in terms of performance, lifetime, and reliability.

## Figures and Tables

**Figure 1 materials-14-05472-f001:**
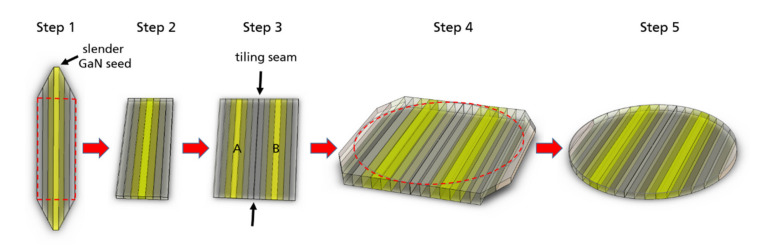
Scheme of the substrate manufacturing steps starting from slender GaN seeds to the finished wafer: Step 1, multi regrowth of a slender GaN seed in the lateral 
$‹11\bar{2}0›$
 directions; three new-grown crystal areas crystallized in three separated processes are marked; Step 2, new crystal sliced from the overgrown seed of Step 1; Step 3, two cut out crystals are joined together by tiling technology; Step 4, GaN continues to grow in the lateral 
$‹11\bar{2}0›$
 directions as well as in vertical 
$\left[000\bar{1}\right]$
 direction in multi regrowth steps; Step 5, when the necessary size is reached the as-grown GaN crystals can be drilled, sliced and polished in a proper way to produce wafers.

**Figure 2 materials-14-05472-f002:**
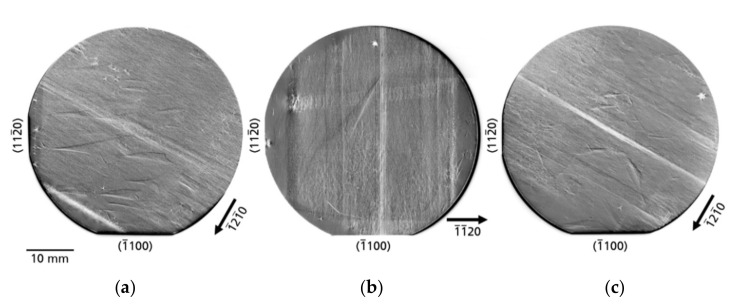
XRT overview topographs of the three investigated substrates: (**a**) substrate A, (**b**) substrate B, and (**c**) substrate C. For the imaging type 
$11\bar{2}0$
 reflections were used.

**Figure 3 materials-14-05472-f003:**
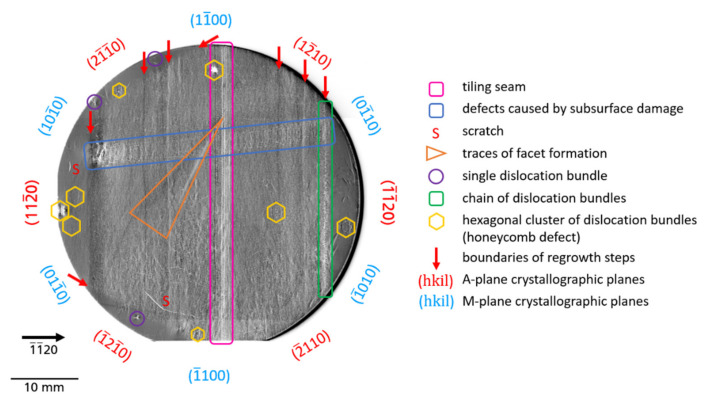
Overview of the typical macroscopic defects found in the Am-GaN 
$\left(0001\right)$
 substrates investigated, using substrate B as an example.

**Figure 4 materials-14-05472-f004:**
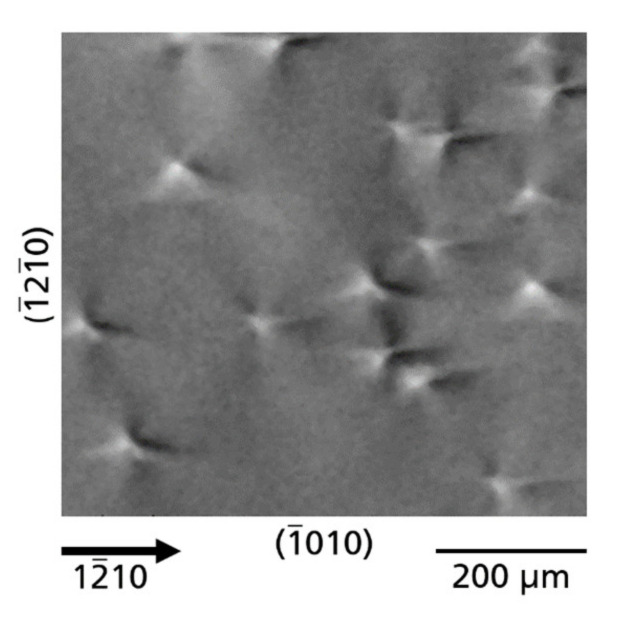
Topograph showing rosette contrasts of TDs with different directions of the bright-dark contrast lobes, indicating different types of TDs (
$1\bar{2}10$
 reflection).

**Figure 5 materials-14-05472-f005:**
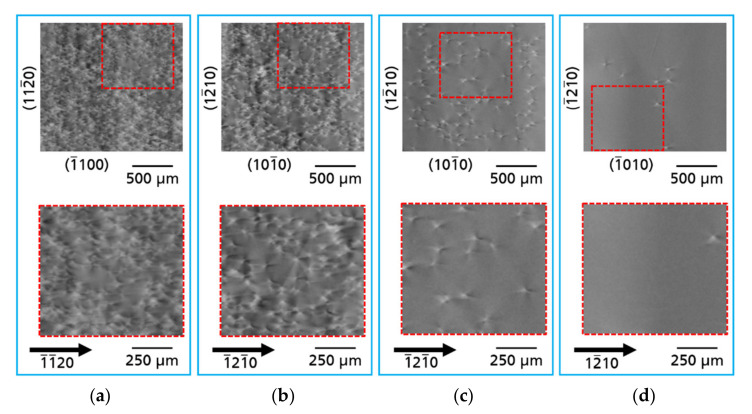
Four areas of threading dislocations in investigated Am-GaN substrates with different TDD: (**a**) high TDD: ~5 × 10^5^ cm^−2^; (**b**) medium TDD: 2 × 10^5^ cm^−2^; (**c**) low TDD: 4 × 10^4^ cm^−2^ and (**d**) ultra-low TDD: 3 × 10^3^ cm^−2^. For the imaging of the topographs type 
$11\bar{2}0$
 reflections were used.

**Figure 6 materials-14-05472-f006:**
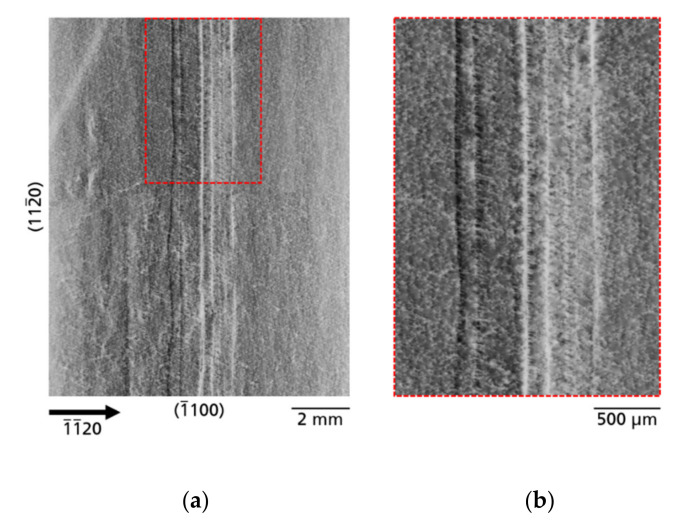
XRT images (
$\bar{1}\bar{1}20$
 reflection) of the tiling seam from sample B: (**a**) six lines of defects are well visible; (**b**) highly enlarged area of the tiling seam.

**Figure 7 materials-14-05472-f007:**
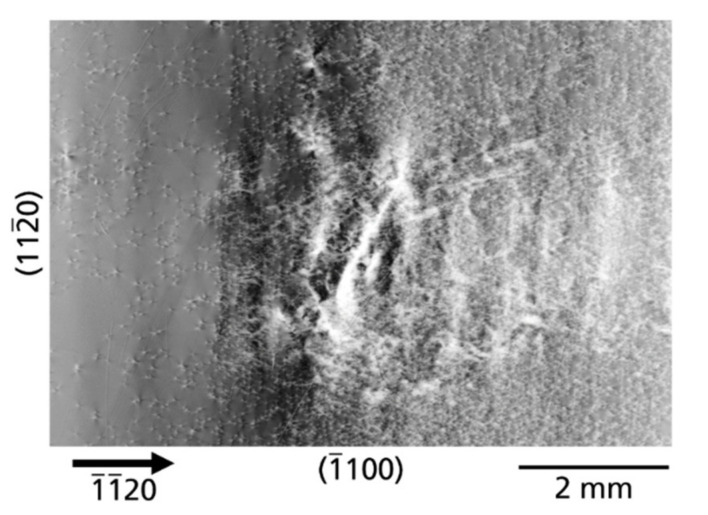
Area with high defect density initiated during growth by subsurface damage of the prepared wafer surface (
$\bar{1}\bar{1}20$
 reflection).

**Figure 8 materials-14-05472-f008:**
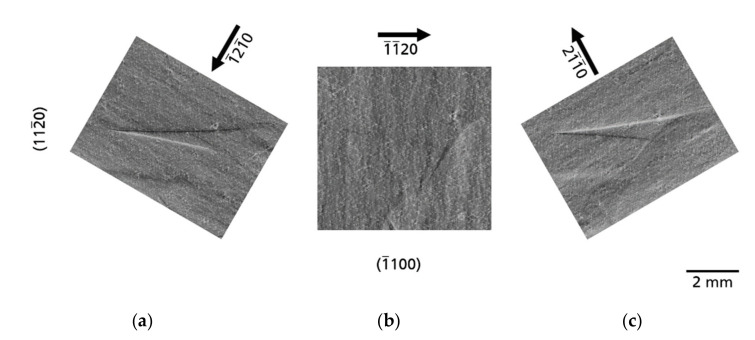
A planar defect caused by facet formation imaged with three different diffraction vectors (sample A); (**a**) reflection 
$\bar{1}\bar{1}20$
, (**b**) reflection 
$\bar{1}2\bar{1}0$
 and (**c**) reflection 
$2\bar{1}\bar{1}0$
. While the defect is clearly visible in the 
$\bar{1}2\bar{1}0$
 and 
$2\bar{1}\bar{1}0$
 reflection topographs, it is almost extinct in the 
$\bar{1}\bar{1}20$
 reflection topograph.

**Figure 9 materials-14-05472-f009:**
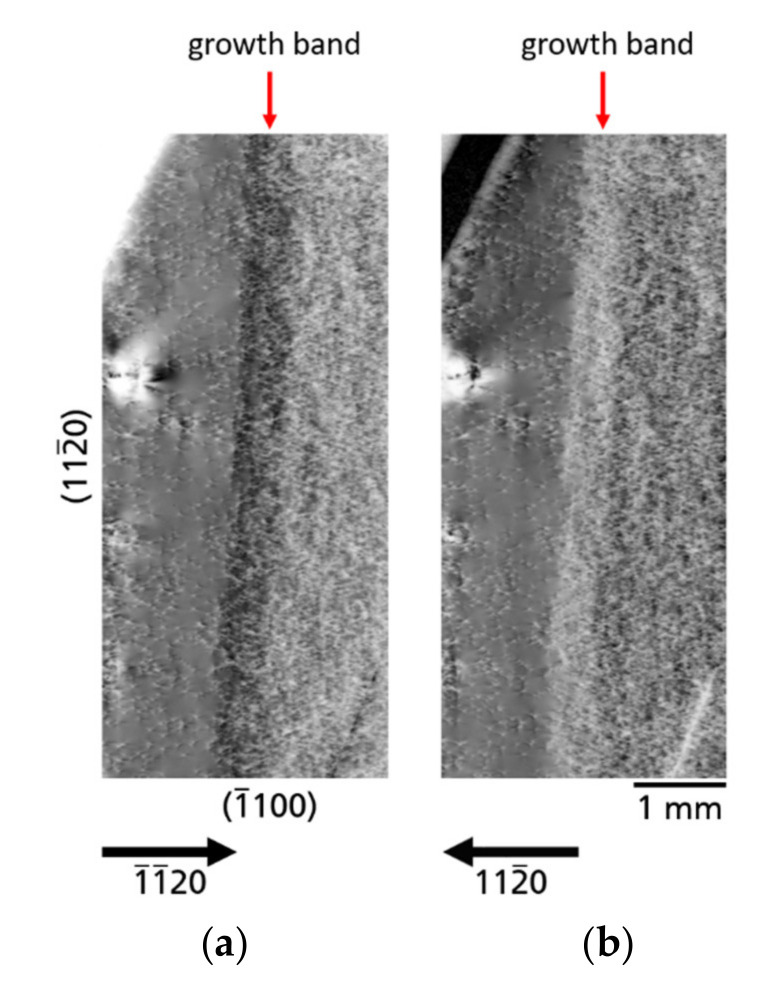
Topographs of the same crystal area of substrate A imaged with diffraction vectors g pointing in opposite directions; (**a**) 
$\bar{1}\bar{1}20$
 reflection and (**b**) 
$11\bar{2}0$
 reflection. The bright-dark contrasts in the growth band area are caused by a slight bending of the lattice plane and are reversed for the used two reflections.

**Figure 10 materials-14-05472-f010:**
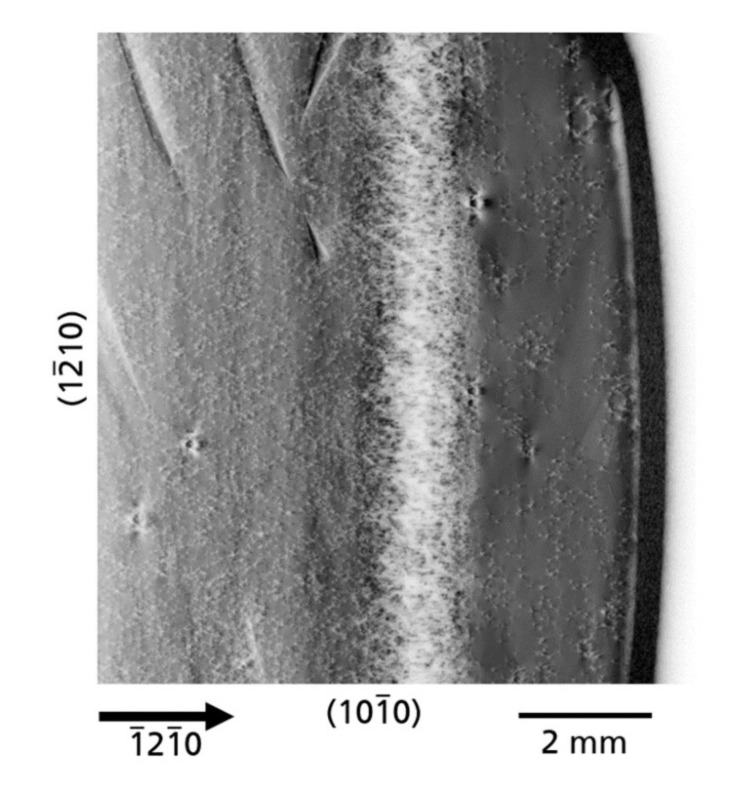
XRT image using 
$\bar{1}2\bar{1}0$
 reflection with a wall of a dense dislocation network observed in crystal A. The Borrmann effect is nearly extinct for this highly defective region.

**Figure 11 materials-14-05472-f011:**
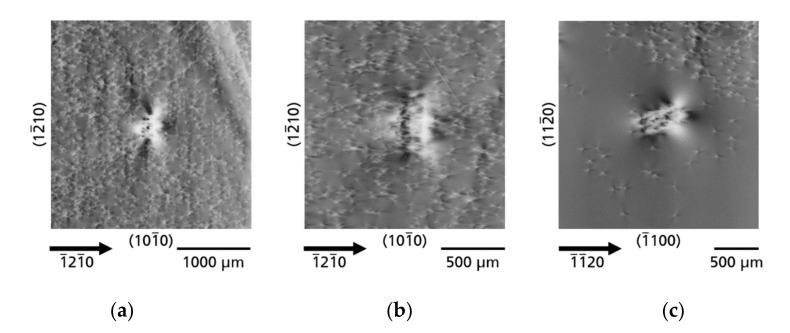
Single dislocation bundles (DBs), which have formed independently of TDD in the surrounding matrix; (**a**) matrix with higher TDD; (**b**) matrix with intermediate TDD; and (**c**) matrix with very low TDD.

**Figure 12 materials-14-05472-f012:**
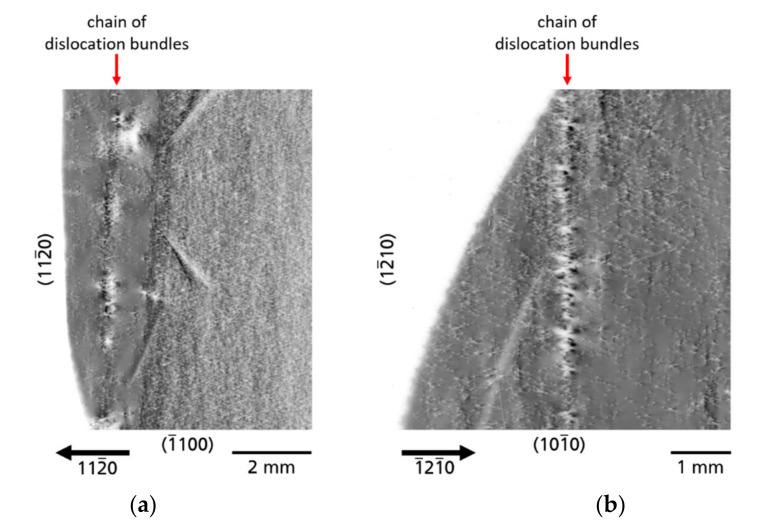
Dislocation bundles (DBs) lined up as a chain along the former growth front; (**a**) “interrupted chain of DBs”; (**b**) “continuous chain of DBs”.

**Figure 13 materials-14-05472-f013:**
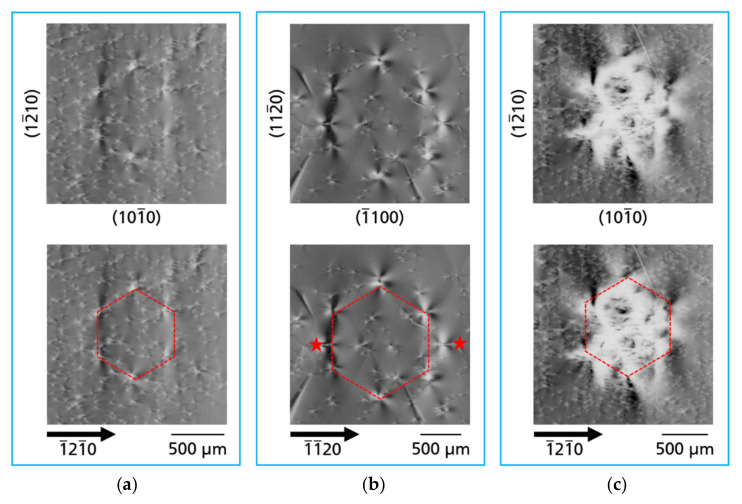
Honeycomb defects formed by a hexagonally arranged cluster of DBs; (**a**) perfect hexagonal honeycomb defect; (**b**) honeycomb defect with clearly formed bright and dark rosette contrasts of the DBs, two additional DBs (marked by stars) nestled into the arrangement of the hexagon; and (**c**) highly distorted variant of the honeycomb defect with distorted inner center showing nearly no Borrmann effect, named cauliflower defect. For the imaging of the topographs type 
$11\bar{2}0$
 reflections were used.

**Figure 14 materials-14-05472-f014:**
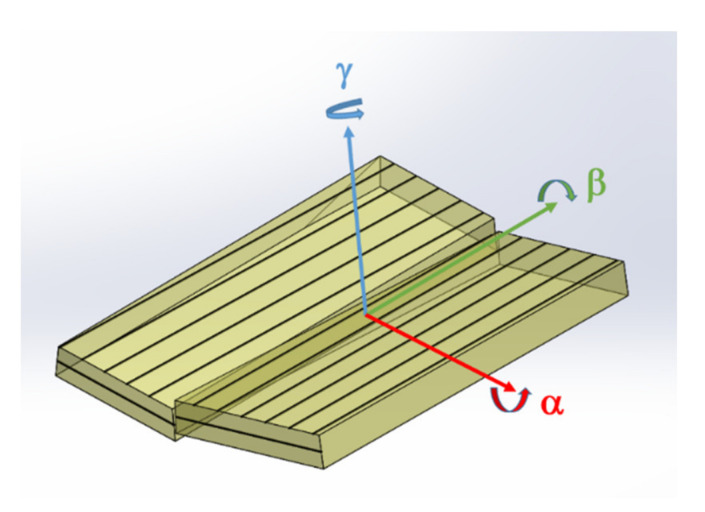
Scheme of tiling technology: combining two crystals to get a one larger; three important misorientation angles α, β, and γ are marked. A mutual misorientation of the seeds in three planes allows obtaining crystals in which the angular differences of α, β and γ are smaller than the variation of crystal off-cut.

## Data Availability

The data presented in this study are available on request from the corresponding author.

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
