# Peer review of "Structural Analysis of Low Defect Ammonothermally Grown GaN Wafers by Borrmann Effect X-ray Topography"

_materials, 2021, doi:10.3390/ma14195472_

Round 1

Reviewer 1 Report

This paper presents a very extensive study of defects in ammonothermally grown GaN wafers using Borrmann X-ray topography. In my judgement the work has been conducted quite carefully and the analysis is very thorough. Such detailed materials characterization work is quite important so that ammonothermal GaN wafers can be matured. I believe that this paper will be of high interest to the GaN community, especially those who use such wafers for GaN epitaxial growth and device fabrication. Based on its depth, quality, and expected interest to the community, I recommend that the paper be accepted for publication. Follow-on work that examines the impact of the observed defects on epitaxy and device performance/reliability will be of interest.

Author Response

Dear Reviewer,

The authors thank you for the constructive review and acceptance for publication.

The English language was proofread by a native speaker and appropriate corrections were made.

Yours sincerely,

Lutz Kirste

Reviewer 2 Report

Peer-reviewed article Kirste et al. presents very important and interesting results on the study of micro and macrodefects in structurally perfect GaN substrates fabricated using the original ammonothermal (Am) method. No less important is the methodological aspect of the article, which clearly demonstrates the capabilities of one of the varieties of X-ray topography based on the Bormann effect. The article was written very carefully and thoroughly, which allowed the authors for the first time to classify many types of possible micro- and macro-defects with an extremely low and inhomogeneously distributed concentration (103-5105 cm-2) in bulk Am-GaN substrates. It is also important that the authors analyze the possible causes of all these defects, which will undoubtedly be taken into account in their future studies. This article is undoubtedly a state-of-the-art both in the field of manufacturing bulk Am-GaN substrates with an extremely low concentration of defects and characterizing their complex structure using Borrmann effect X-ray topography. Moreover, this method, as shown by a comparison of this study with the previous work of the authors [Ref.11], significantly expands the understanding of the formation of defects in Am-GaN substrates. It will be of great interest to a wide range of specialists working in the field of manufacturing bulk substrates, their structural characterization in order to use them in the manufacture of semiconductor devices.

Meanwhile, while reading the article, I had a question that requires clarification. Is it right that the four areas with the different TD concentration, shown in Fig.5,  were measured in the different samples A,B,C at the various distances from the tiling seams separated the substrates to the two halves. If yes, it would be desirable to give these approximate distances in the figures or in their captions.

Author Response

Dear Reviewer,

The authors thank you for the constructive review and acceptance for publication.

The dislocation densities shown in Fig. 5 exemplify the areas with different dislocation densities in the three substrates studied. It is also true that in each case there is a trend from higher dislocation density starting in the area of the tile seam to ultra low dislocation density in the edge areas of the substrates. However, the authors shy away from quantifying distances for the respective dislocation density regimes. Even though the trend is observed for each substrate, however, there are notable differences from crystal to crystal for the same spacing. This is probably related to factors such as the duration of the individual growth cycles and others. The quantification requested by the referee could be misleading and with the referee's agreement the authors would like to omit it.

The English language was proofread by a native speaker and appropriate corrections were made.

Yours sincerely,

Lutz Kirste

Reviewer 3 Report

The authors have presented the detailed structural analysis study of GaN wafers by X-ray topography using the Borrmann effect and have provided a good introduction to the manuscript. The reviewer also agrees that this may be the first report of the Borrmann effect for GaN wafer substrates, which is of great interest for the wide bandgap-based electronic/photonic materials/devices research community. The structural characterization and results generally support the thesis of the manuscript. However, it would appreciate if the authors could compare the threading dislocation density information of this work with existing other reports in a table. Overall, the current manuscript can be accepted after minor revision. 

Author Response

Dear Reviewer,

The authors thank you for the constructive review and acceptance for publication.

The report proposes the preparation of a comparative table for the dislocation densities estimated from this work and the literature. While writing this paper, the authors also had the idea of such a table. This was finally not done, because the authors think that this comparison is difficult. There are no data from comparable measurement conditions in the literature for a fair comparison. E.g., HVPE-GaN grown on foreign substrate shows a TDD larger than 106 cm-2 (see reference [43, 37]. However, these dislocation densities are from other methods. Am-GaN was measured e.g. by Ref. [12] using grazing incidence SWB-XRT. However, this method is not volume sensitive. For this reason, the authors decided to compare to other GaN substrates by the presence of the Borrmann effect, see Appendix A. Although this comparison is not quantitative, it shows on a qualitative level the clear difference in GaN crystal perfection.

The English language was proofread by a native speaker and appropriate corrections were made.

Yours sincerely,

Lutz Kirste